# Knowledge and attitude towards mpox: Systematic review and meta-analysis

Darwin A. León-Figueroa[1], Joshuan J. Barboza[2], Abdelmonem Siddiq[3], Ranjit Sah[4,5,6], Mario J. Valladares-Garrido[7,8]*, Alfonso J. Rodriguez-Morales[9,10]

**1** Facultad de Medicina Humana, Universidad de San Martín de Porres, Chiclayo, Peru, **2** Facultad de Ciencias de la Salud, Escuela de Medicina, Universidad César Vallejo, Trujillo, Peru, **3** Faculty of Pharmacy, Mansoura University, Mansoura, Egypt, **4** Department of Microbiology, Tribhuvan University Teaching Hospital, Institute of Medicine, Kathmandu, Nepal, **5** Department of Microbiology, Dr. D. Y. Patil Medical College, Hospital and Research Centre, Dr. D. Y. Patil Vidyapeeth, Pune, Maharashtra, India, **6** Department of Public Health Dentistry, Dr. D.Y. Patil Dental College and Hospital, Dr. D.Y. Patil Vidyapeeth, Pune, Maharashtra, India, **7** Universidad Continental, Lima, Peru, **8** Oficina de Epidemiología, Hospital Regional Lambayeque, Chiclayo, Peru, **9** Master of Clinical Epidemiology and Biostatistics, Universidad Cientifica del Sur, Lima, Peru, **10** Gilbert and Rose-Marie Chagoury School of Medicine, Lebanese American University, Beirut, Lebanon

* mvalladares@continental.edu.pe

**Data Availability Statement:** If the data are all contained within the manuscript and/or Supporting Information files, enter the following: All relevant

## Abstract

### Background

The increase in mpox incidence underscores the crucial need to understand and effectively address prevention, early detection, and agile response to this disease. Therefore, the present study aims to determine the knowledge and attitude towards mpox.

### Methods

A systematic review and comprehensive literature meta-analysis were conducted using prominent databases such as PubMed, Scopus, Web of Science, Embase, and ScienceDirect, with an updated search until June 25, 2023. The quality of the included observational studies was assessed using the Joanna Briggs Institute's Statistical Meta-Analysis Review Instrument. The collected data were recorded in a Microsoft Excel spreadsheet, and analyses were conducted using R software version 4.2.3. Additionally, Cochran's Q statistics were applied to assess the heterogeneity of the included studies.

### Results

A total of 299 articles were retrieved from 5 databases. This study included 27 cross-sectional articles with a total sample of 22,327 participants, of which 57.13% were women. The studies were conducted in 15 countries through an online survey. All studies had a moderate level of quality. The combined prevalence of a good level of knowledge about mpox was 33% (95% CI: 22%-45%; 22,327 participants; 27 studies; $I^2 = 100\%$), and the combined prevalence of a positive attitude towards mpox was 40% (95% CI: 19%-62%; 2,979 participants; 6 studies; $I^2 = 99\%$). Additionally, as a secondary outcome, the combined prevalence of the intention to vaccinate against mpox was 58% (95% CI: 37%-78%; 2,932 participants; 7 studies; $I^2 = 99\%$).

data are within the manuscript and its Supporting Information files.

**Funding:** The author(s) received no specific funding for this work.

**Competing interests:** The authors have declared that no competing interests exist.

## Conclusion

Good knowledge and a positive attitude towards mpox were found to be low. The findings of this study highlight the need to identify gaps and focus on implementing educational programs on mpox.

## Terms used

Joanna Briggs Institute Meta-Analysis of Statistics Assessment and Review Instrument (JBI-MAStARI), Prospective International Registry of Systematic Reviews (PROSPERO), and Preferred Reporting Items for Systematic Reviews and Meta-Analyses (PRISMA)

## 1. Introduction

The spread of mpox in humans has raised significant concerns in various countries globally, extending beyond Africa, especially in a context marked by the COVID-19 pandemic [1]. As of June 30, 2024, 97,962 cases of mpox have been reported in 118 countries, 111 of which have not historically reported it [2].

Mpox is a reemerging zoonotic viral disease caused by the mpox virus, an Orthopoxvirus from the Poxviridae family [3]. Individuals affected by mpox experience a time interval of 7 to 21 days between exposure and the onset of distinctive clinical symptoms, which include fever, headache, muscle pain, back pain, chills, skin rash, and lymphadenopathy [4].

The spread of the mpox virus from one person to another extends beyond close direct contact [5]. Given the rapid development of the mpox virus, how it spreads includes skin wounds, genital lesions, throat secretions, seminal fluid, and blood [6, 7]. The rapid expansion of the outbreak in 2022 has raised concerns, mainly because over 95% of the cases affected men who have sex with other men [8, 9].

In the face of a public health emergency like the mpox outbreak in 2022, countries must have immediate action plans to prevent diseases promptly. Additionally, providing prevention equipment and disseminating clear information about the signs and symptoms of the disease among the general population is essential [10, 11]. It is also vital to ensure that healthcare professionals receive ongoing training or participate in short-term emergency training programs related to mpox [10, 12]. These measures will effectively contribute to mitigating the impact of mpox [11].

Despite efforts and thorough planning, several challenges persist that need to be overcome, including the deficiency in the level of education and the limited knowledge of citizens regarding the current health crisis [13, 14]. Therefore, this study aims to determine the combined prevalence of knowledge and attitudes regarding mpox.

## 2. Materials and methods

### 2.1. Protocol and registration

This systematic review and meta-analysis were conducted following the guidelines of the PRISMA checklist [15] (**S1 Table**). The protocol for this research has been appropriately registered in PROSPERO (**CRD42023439782**), ensuring transparency and rigor in the process.

## 2.2. Eligibility criteria

**Inclusion criteria.**   All observational studies on the prevalence of knowledge, attitude, or both, regarding mpox were considered. No restrictions were imposed regarding gender, health status, language, time, quality, or geographic location. However, only those studies that were available in their entirety provided sample size information presented data related to any aspect of knowledge and attitude towards mpox or provided data from which the required results could be calculated, were included.

**Exclusion criteria.**   The following studies were excluded: those containing duplicate information, those whose research topics were unrelated to the objective of our study, as well as those using a design different from an observational study. Additionally, articles lacking full text were discarded due to insufficient data or not reporting the desired results.

## 2.3. Information sources and search strategy

Three expert researchers conducted exhaustive searches in various databases, including PubMed, Scopus, Embase, Web of Science, and ScienceDirect. Keywords such as "mpox," "knowledge," "awareness," and "attitude" were used as part of the search strategy. Specific search strategies for each database are detailed in **S2 Table**. The initial search was conducted on June 1, 2023, and subsequently updated on June 25, 2023.

## 2.4. Study selection

The search strategy results were stored and managed using the Endnote software. After eliminating duplicate articles, three experts independently conducted a preliminary selection of the remaining articles by reading the titles and abstracts, following predefined criteria. Subsequently, two other researchers thoroughly reviewed the complete reports to determine if they met the inclusion criteria. Any discrepancies would be resolved through discussions and consultations with a sixth investigator.

## 2.5. Main and secondary results of the study

This study addressed two main aspects related to knowledge and attitudes towards mpox.

**Knowledge about mpox.**   The knowledge base of the participants in this study relied on the reports of the included articles, which revealed either good general knowledge or a high level of specific knowledge about mpox. The criteria used to determine the combined prevalence of knowledge covered modes of transmission, clinical symptoms, treatment, prevention, and the diagnosis of mpox.

**Attitude towards mpox.**   The participants' attitude in this study was based on the analysis of the included articles, which encouraged a positive attitude towards mpox. Positive attitudes toward mpox included confidence in the overall ability to control the epidemic, in the effectiveness of preventive and control measures, and in the perception that health actions are adequate to prevent its spread.

**Intention to vaccinate against mpox.**   The intention of the participants to vaccinate against mpox in this study was based on the analysis of the included articles, which reported the importance of getting vaccinated against mpox if the vaccine was available or as a preventive measure.

## 2.6. Quality assessment

Three independent authors evaluated the quality of the included studies using the "JBI-MAStARI" method for observational studies. In the event of any discrepancies among the

evaluators, a fourth author intervened to address and resolve them. To perform this evaluation, a checklist composed of eight critical parameters was used to assess the responses as "yes," "no," "unclear," or "not applicable." The quality of the studies was classified based on their score as high (≥7 points), moderate (4 to 6 points), or low (<4 points) [16] (**S3 Table**).

### 2.7. Data collection process and data items

Two independent researchers meticulously collected relevant data from the selected articles. The following details were extracted and recorded in an Excel spreadsheet: first author's name, publication year, country, sample size, study population, gender (male and female), the prevalence of mpox knowledge, prevalence of attitudes towards mpox, number of cases with knowledge of mpox, and number of cases with attitudes towards mpox. Finally, a third researcher verified the extracted data to ensure accuracy and eliminate incorrect information.

### 2.8. Data analysis

Firstly, the selected articles were entered into a Microsoft Excel spreadsheet to perform the analysis using R, version 4.2.3. Narrative tables and charts were used to present the research results. To estimate the joint prevalence of mpox knowledge and attitudes, an inverse variance-weighted random-effects model was used. This technique, commonly used in meta-analyses, allows the results of several independent studies to be combined. The model takes into account both within-study variability (within-study variance) and between-study variability (between-study variance) [17]. The Cochrane Q statistic was employed to assess heterogeneity among studies and quantified using the $I^2$ index, where 25%, 50%, and 75% indicated low, moderate, and high heterogeneity, respectively [18]. Funnel plots and Egger's regression test were used to check for publication bias. Publication bias occurs when the results of published studies are not representative of all studies conducted, usually because studies with non-significant results are less likely to be published [19]. A possible publication bias was considered when the p-value was < 0.05 [20].

Subgroup analyses were performed according to study population and country. A forest plot was used to illustrate the combined prevalence of good knowledge and attitudes towards mpox, including 95% confidence intervals.

## 3. Results

### 3.1. Study selection

A total of 299 articles were retrieved from 5 databases. After removing duplicates (n = 125), researchers analyzed 174 articles. Then, the titles and abstracts of these articles were reviewed, and 54 were selected for a thorough full-text review. Finally, 27 articles were included in the study [21–47]. The PRISMA flow diagram shows the study selection process (**Fig 1**).

### 3.2. Characteristics of the included studies

This study included 27 cross-sectional articles with a total sample size of 22,327 participants (**Table 1**). The sample composition consisted of 42.83% males, 57.13% females, and 0.04% others (undefined or unreported) [21–47]. The studies were conducted in 15 countries using an online survey, where questionnaires were sent via e-mail and other communication channels to those with Internet access. The articles cover the period between 2020 and 2023 in their publication year. The sample sizes ranged from 111 to 5,874. Regarding the prevalence of knowledge and attitude towards mpox, the ranges observed were from 0.6% to 65.46% and 12.2% to 84.83%, respectively [21–47].

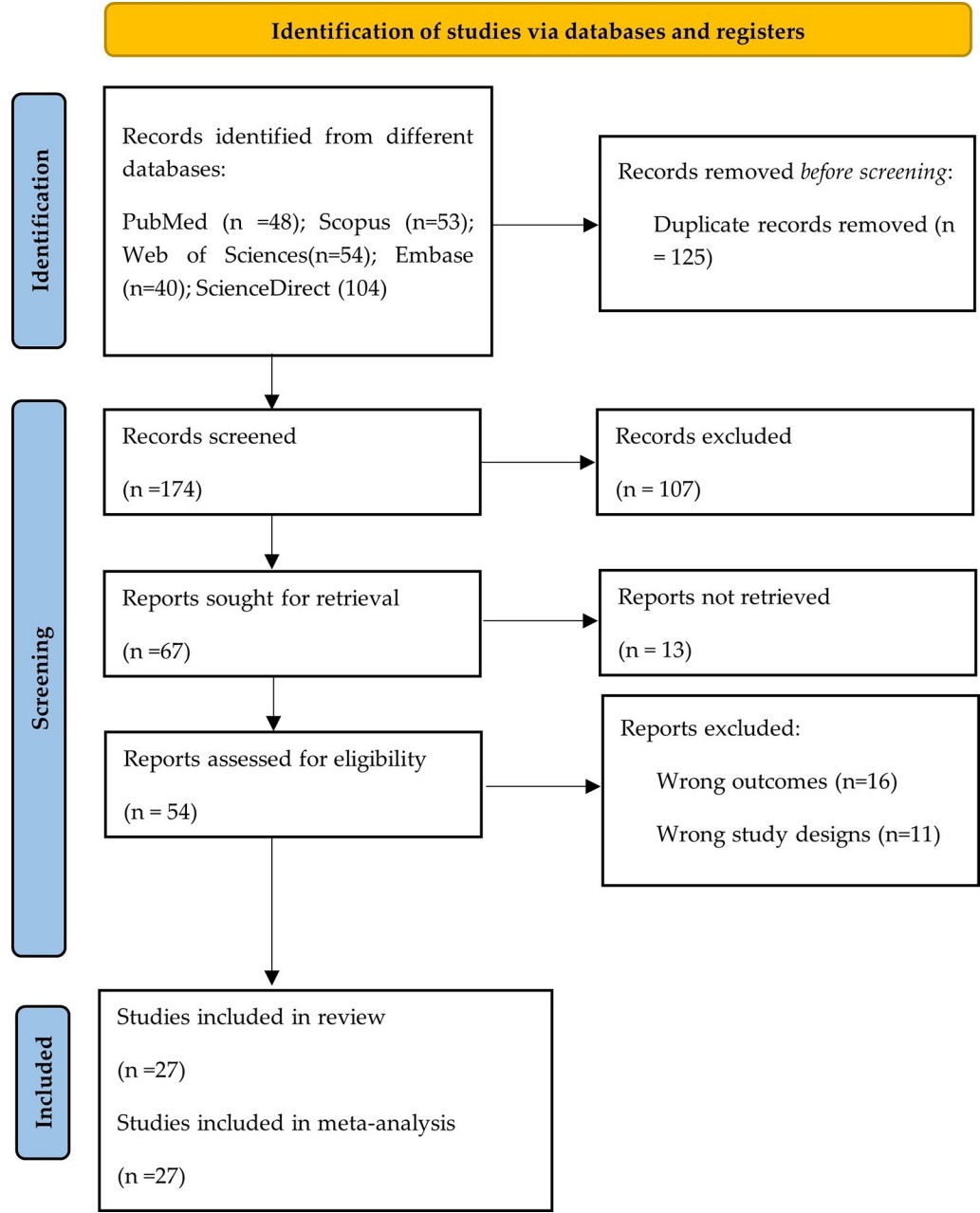

**Fig 1. Research selection procedure based on the PRISMA flowchart.**

## 3.3. Quality of the included studies and publication bias

The studies were evaluated regarding quality using the JBI-MAStARI for observational research. It was determined that all studies had a moderate level of quality [21–47] (**S3 Table**). We examined the publication bias of articles that reported the level of good knowledge and positive attitude about mpox (**S1 Fig**). Egger's test, applied to assess publication bias in studies related to the level of good knowledge about mpox, revealed a value of p = 0.0135 (t = 2.66, df = 25), leading to the rejection of the null hypothesis of symmetry. This finding suggests the possible presence of publication bias in the studies analyzed (**S1 Fig**) [21–47]. Articles

**Table 1. Characteristics of the included studies on knowledge and attitudes toward mpox.**

| Authors | Year | Study design | Country | Sample size (n) | Study population | Sex | | Prevalence of good Knowledge | Prevalence of positive attitude | Number of study participant with good knowledge | Number of study participant with positive attitude | Vaccination intention (n, %) |
|---|---|---|---|---|---|---|---|---|---|---|---|---|
| | | | | | | M | F | | | | | |
| Hasan M, et al. [21] | 2023 | Cross sectional | Bangladesh | 389 | Physicians | 184 | 205 | 30.59% | 84.83% | 119 | 330 | NS |
| Swed S, et al. [22] | 2023 | Cross sectional | Arabic regions | 5874 | Healthcare Workers | 2455 | 3419 | 0.6% | NS | 36 | NS | NS |
| Dong C, et al. [24] | 2023 | Cross sectional | China | 521 | General population | 264 | 257 | 59.3% | NS | 309 | NS | 398 (76.40%) |
| Swed S, et al. [23] | 2023 | Cross sectional | Arabic regions | 3665 | General population | 1477 | 2187 | 45.5% | NS | 1668 | NS | NS |
| Peng X, et al. [25] | 2023 | Cross sectional | China | 639 | Physicians | 208 | 431 | 65.46% | NS | 418 | NS | NS |
| Berdida DJE, et al. [26] | 2023 | Cross sectional | Philippines | 575 | General population | 179 | 396 | 4.87% | NS | 28 | NS | NS |
| Elkhwesky Z, et al. [27] | 2023 | Cross sectional | Egypt | 453 | Hotel employees | 363 | 90 | 18.1% | NS | 82 | NS | NS |
| Youssef D, et al. [28] | 2023 | Cross sectional | Lebanon | 793 | General population | 231 | 562 | 33.04% | NS | 262 | NS | NS |
| Malaeb D, et al. [29] | 2023 | Cross sectional | Lebanon | 646 | Healthcare Workers | 302 | 344 | 33.7% | 30.7% | 218 | 198 | NS |
| Chen Y, et al. [30] | 2023 | Cross sectional | China | 154 | Male sex workers | 154* | 0 | 49.4% | NS | 74 | NS | 97 (63%) |
| Lounis M, et al. [31] | 2023 | Cross sectional | Algeria | 111 | Healthcare Workers | 33 | 78 | 64.9% | NS | 72 | NS | 43 (38.7%) |
| Ahmed SK, et al. [32] | 2023 | Cross sectional | Iraq | 510 | General population | 277 | 233 | 40.4% | 12.2% | 206 | 62 | 132 (25.9%) |
| Das SK, et al. [33] | 2023 | Cross sectional | Nepal | 205 | Healthcare Workers | 114 | 91 | 60.40% | 51.70% | 124 | 106 | NS |
| Al-Mustapha AI, et al. [34] | 2023 | Cross sectional | Nigeria | 822 | General population | 472 | 342 | 52.19% | NS | 429 | NS | NS |
| Ren F, et al. [35] | 2023 | Cross sectional | China | 1028 | General population | 326 | 702 | 56.5% | NS | 581 | NS | NS |
| Alrasheedy AA, et al. [36] | 2023 | Cross sectional | Saudi Arabia | 189 | Community Pharmacists | 164 | 25 | 31.2% | NS | 59 | NS | NS |
| Sahin TK, et al. [37] | 2022 | Cross sectional | Turkey | 283 | Physicians | 117 | 166 | 32.5% | 41.7% | 92 | 118 | 89 (31.4%) |
| Alshahrani NZ, et al. [38] | 2022 | Cross sectional | Saudi Arabia | 314 | Medical Students | 131 | 183 | 28% | NS | 88 | NS | NS |
| Jairoun AA, et al. [41] | 2022 | Cross sectional | United Arab Emirates | 558 | University students | 208 | 350 | 22.8% | NS | 127 | NS | NS |
| Kaur A, et al. [42] | 2022 | Cross sectional | India | 410 | Dental professionals | 232 | 178 | 28% | NS | 116 | NS | NS |
| Alshahrani NZ, et al. [39] | 2022 | Cross sectional | Saudi Arabia | 480 | General population | 198 | 282 | 48% | NS | 228 | NS | NS |
| Kumar N, et al. [43] | 2022 | Cross sectional | Pakistan | 946 | University students | 432 | 514 | 6.3% | 20.5% | 60 | 194 | 640 (67.7%) |

*(Continued)*

**Table 1.** (Continued)

| Authors | Year | Study design | Country | Sample size (n) | Study population | Sex | | Prevalence of good Knowledge | Prevalence of positive attitude | Number of study participant with good knowledge | Number of study participant with positive attitude | Vaccination intention (n, %) |
|---|---|---|---|---|---|---|---|---|---|---|---|---|
| | | | | | | M | F | | | | | |
| Ajman F, et al. [44] | 2022 | Cross sectional | Saudi Arabia | 1130 | Healthcare Workers | 422 | 708 | 23.2% | NS | 261 | NS | NS |
| Alshahrani NZ, et al. [40] | 2022 | Cross sectional | Saudi Arabia | 398 | Physicians | 226 | 172 | 55% | NS | 219 | NS | NS |
| Harapan H, et al. [45] | 2020 | Cross sectional | Indonesia | 432 | Physicians | 140 | 292 | 36.5% | NS | 158 | NS | NS |
| Harapan H, et al. [46] | 2020 | Cross sectional | Indonesia | 407 | Physicians | 128 | 279 | 9.3% | NS | 38 | NS | 381 (93.6%) |
| Harapan H, et al. [47] | 2020 | Cross sectional | Indonesia | 395 | Physicians | 125 | 270 | 9.4% | NS | 37 | NS | NS |

M/F: Male/Female; NS: Not specified

*MSM: men who have sex with men.

reporting a positive attitude towards mpox were not assessed for publication bias, as there were fewer than ten studies.

### 3.4. Level of knowledge of and attitude towards mpox

The aggregated prevalence and 95% confidence interval of knowledge and attitudes towards mpox among study participants are presented in a forest plot (**Figs 2 and 3**) [21–47]. The random-effects model showed that the combined level of good knowledge about mpox was 33% (95% CI: 22%–45%; 22,327 participants; 27 studies; $I^2$ = 100%; p < 0.01) (**Fig 2**) [21–47]. The estimated overall positive attitude towards mpox was 40% (95% CI: 19%–62%; 2,979 participants; 6 studies; $I^2$ = 99%; p < 0.01) [21, 29, 32, 33, 37, 43] (**Fig 3**).

### 3.5. Secondary outcomes

The pooled prevalence of intention to vaccinate against mpox was 58% (95% CI: 37%–78%; 2,932 participants; 7 studies; $I^2$ = 99%; p < 0.01) [24, 30–32, 37, 43, 46] (**S2 Fig**).

### 3.6. Subgroup analysis

Subgroup analyses were performed based on the study region and study population [21–47].

**3.6.1. Subgroup analysis by study region.** A subgroup analysis was performed based on country. The pooled prevalence of a high level of knowledge about mpox was found to be higher in Algeria (65%, 95% CI: 56%, 74%) [31] and lower in the Philippines (5%, 95% CI: 3%, 7%) [26] (**S3 Fig**). However, the pooled prevalence of a positive attitude towards mpox was higher in Bangladesh (85%, 95% CI: 81%, 89%) [21] and lower in Iraq (12%, 95% CI: 9%, 15%) [32] (**S4 Fig**).

**3.6.2. Subgroup analysis by study population.** A subgroup analysis based on the study population was conducted and divided into two groups: healthcare personnel (doctors, medical students, dental health professionals, and healthcare employees) and the general population (hotel workers, sex workers, university students from different health-related disciplines, and participants from the public). The overall prevalence of a high level of knowledge about mpox was higher in the general population (34%; 95% CI: 23%–46%; 10,505 participants; 12 studies;

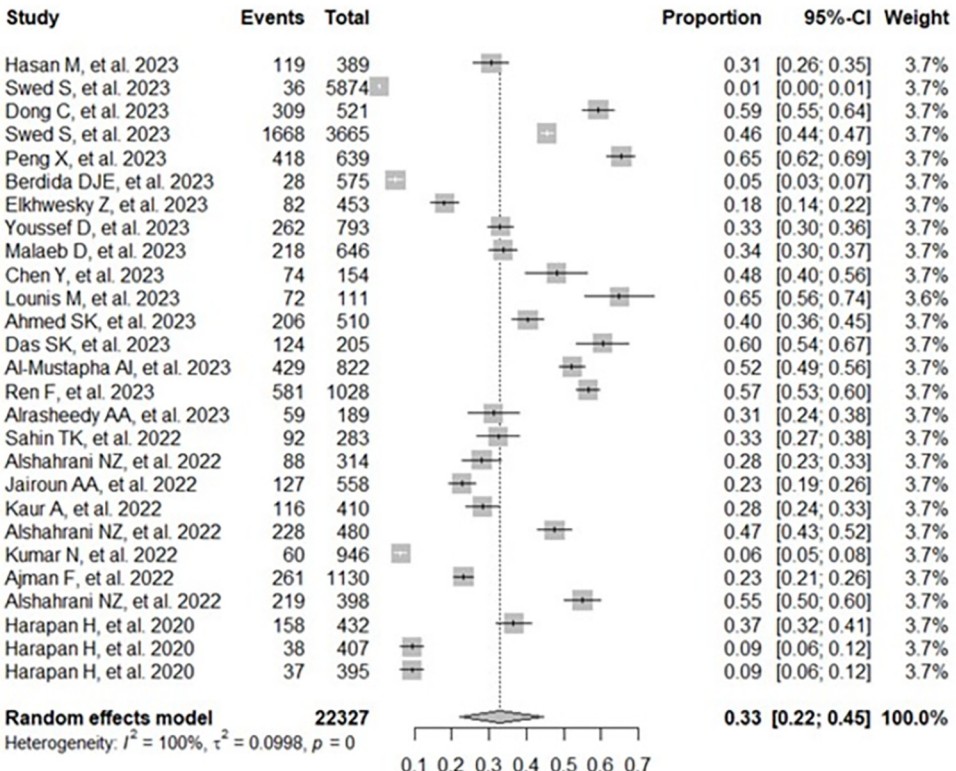

**Fig 2. Forest diagram representing the combined prevalence of good knowledge about monkeypox among study participants.**

$I^2$ = 99%; p < 0.01) [23, 24, 26–28, 30, 32, 34, 35, 39, 41, 43] and lower among healthcare personnel (32%; 95% CI: 17%– 49%; 11,822 participants; 15 studies; $I^2$ = 100%; p < 0.01) [21, 22, 25, 29, 31, 33, 36–38, 40, 43–47] (**S5 Fig**). The overall prevalence of a positive attitude towards mpox was higher among healthcare personnel (53%; 95% CI: 26%– 79%; 1,523 participants; 4 studies; $I^2$ = 99%; p < 0.01) [21, 29, 33, 37] and lower in the general population (16%; 95% CI: 9%– 25%;1,456 participants; 2 studies; $I^2$ = 94%; p < 0.01) [32, 43] (**S6 Fig**).

## 4. Discussion

Mpox is not currently considered a public health emergency of international concern; however, it continues to be transmitted in several countries. A thorough understanding of prevention and control measures for this disease is essential.

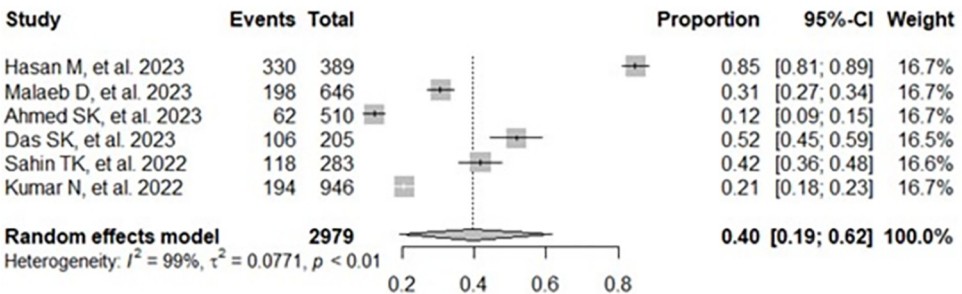

**Fig 3. Forest diagram representing the combined prevalence of positive attitudes.**

Given the diversity of research on mpox, multiple studies have been conducted to assess the knowledge of different population targets about mpox to understand the gap of knowledge that can be covered by appropriate educational tools to increase the knowledge score about this disease and to curb its transmission by following the infection control measures [21, 22, 24, 48]. So we have conducted a systematic review and meta-analysis to gain a more comprehensive understanding of the prevalence of good knowledge levels and attitudes of the general population towards mpox. All relevant studies found through various online search engines were considered to achieve this. Additionally, a subgroup analysis was performed to examine the prevalence of individuals with positive attitudes and a good level of knowledge about mpox based on the region of study and the population analyzed.

The findings of this study revealed that the combined prevalence of good knowledge about mpox was 33%, and the subgroup analysis revealed a total prevalence of good knowledge among the general population and healthcare personnel equal to 34% and 32%, respectively. A meta-analysis by Jahromi AS et al., which included 22 studies involving 27,731 health care workers, revealed that 26% of them had a good knowledge of mpox [49]. To improve knowledge about mpox, it is crucial to identify reliable and up-to-date sources of information that provide accurate data on its transmission, symptoms, prevention, and treatment. This is essential to effectively understanding and addressing this disease, with a strong emphasis on online sources such as social networks (59%) and the Internet (61%) [50].

Another result of this study was the combined prevalence of positive attitudes towards mpox, which was 40%. The subgroup analysis results indicated that the combined proportion of positive attitudes towards mpox among healthcare workers was 53%, and the general population was 16%. Different research has assessed this positive attitude towards mpox, ranging from 12% to 85% [21, 29, 32, 33, 37, 43]. A meta-analysis by Jahromi AS et al., which included six studies with 14,388 health care workers, revealed that 34.6% of them had a positive attitude toward mpox [49]. That could be explained by the disparity in how different populations respond to disease severity and adopt protective measures, which may be attributed to socioeconomic, cultural, information access, and distrust in the healthcare system or government policies. It is essential to address these factors to ensure a more equitable and effective response to any disease and promote the adoption of public health measures to benefit the entire population [51, 52].

The difference in the prevalence estimates of the knowledge and the attitude score between our findings and other individual papers may be justified by the difference in the culture of the population from one country to another, the difference in the target group (general population, medical students and healthcare workers), the difference in the timeline at which each study was conducted, the difference in the survey methods used in the assessment of the knowledge and the attitude, and other factors that should be taken into our consideration.

The secondary outcome revealed that the prevalence of the intention to vaccinate against mpox was 58%, consistent with a meta-analysis conducted by Ulloque-Badaracco et al., which reported that the prevalence of mpox vaccine acceptance equals 56% [53]. Another meta-analysis proposed by León-Figueroa DA et al., which included 29 articles with a total sample of 52,658 participants, determined that the combined prevalence of intention to be vaccinated against mpox was 61% [54]. It was recommended by the World Health Organization (WHO) and The Center for Disease Control and Prevention (CDC) to vaccinate certain groups of the population who are at risk of developing mpox in terms of pre-exposure and postexposure prophylaxis using JYNNEOS, ACAM2000, and LC16m8 vaccines [55–57]. However, there is variation in the prevalence of intention to vaccinate against mpox reported in the individual studies; this variation could be due to fear of unknown adverse reactions and doubts about the efficacy and safety of the mpox vaccine [54, 58].

We recommend that further research is needed to cover the knowledge of the different population groups, including the general public, healthcare professionals, and students, regarding mpox disease, especially in countries with missing data. Further research is needed in countries with many mpox-infected patients at different intervals to track the change in the people's knowledge, attitude, intention to get vaccinated, and their maintenance on tracking the infection control measures.

This research has several limitations. First, the use of self-reported questionnaires could introduce biases, as participants could provide socially acceptable answers or exaggerate their knowledge about mpox, thus affecting the validity of the data. To mitigate this problem, it is crucial to recommend the use of proxy questions, ensure the anonymity of responses, and perform consistency analysis. Second, the high heterogeneity among the included studies ($I^2 >$ 75%) indicates diversity in the methodologies and populations investigated, which could limit the generalizability of the findings. Our study addressed this issue through subgroup analysis, clear inclusion criteria, and the use of random-effects models, thus providing more accurate and robust data. Third, we found evidence of publication bias, which we addressed using tests such as Egger's test and funnel plots. Fourth, variations in questionnaire design, distribution methods, and participant demographics could introduce confounding factors into the analysis. Despite these differences, the studies presented similar general criteria regarding transmission, clinical symptoms, diagnosis, treatment, and prevention of mpox. Finally, variability in outcomes could be attributed to sociodemographic, economic, and cultural factors, as well as access to education and trust in the health system or government policies of each country.

Nevertheless, this research has strengths. First, an exhaustive search was carried out in multiple databases without language restrictions, which increased the completeness of the review. Second, robust tools were used to assess quality, and statistical analysis was performed, which reinforced the validity of the results (JBI-MAStARI, PRISMA, Egger's test, funnel plots, and R software). Third, article selection and data extraction were performed independently by more than three investigators. Finally, this study represents the first systematic review and meta-analysis assessing the prevalence of good knowledge and positive attitudes toward mpox, providing reliable data that can be used by policymakers to improve knowledge and attitudes toward mpox.

## 5. Conclusions

In conclusion, this systematic review and meta-analysis reported a significant gap in good knowledge and positive attitudes towards mpox. Furthermore, the combined prevalence of good knowledge and positive attitudes differed across study populations, regions, and publication years. A holistic and multisectoral approach is necessary for the successful understanding of mpox. Additional healthcare education and communication are crucial for improving knowledge and attitudes regarding mpox.

## Supporting information

**S1 Table. PRISMA checklist (PRISMA 2020 main checklist and PRIMSA abstract checklist).**
(DOCX)

**S2 Table. The adjusted search terms as per searched electronic databases.**
(DOCX)

**S3 Table. Quality of the included studies.**
(DOCX)

**S1 Fig. Funnel plot and Egger's test illustrate the publication bias of the included studies.**
(TIF)

**S2 Fig. Forest plot showing prevalence of intention to vaccinate against mpox among study participants.**
(TIF)

**S3 Fig. Subgroup analysis by country on good knowledge of mpox.**
(TIF)

**S4 Fig. Subgroup analysis by country on positive attitude toward mpox.**
(TIF)

**S5 Fig. Subgroup analysis by study subjects on good knowledge of mpox.**
(TIF)

**S6 Fig. Subgroup analysis by study subjects on positive attitude toward mpox.**
(TIF)

## Author Contributions

**Conceptualization:** Darwin A. León-Figueroa, Joshuan J. Barboza, Abdelmonem Siddiq, Ranjit Sah, Mario J. Valladares-Garrido, Alfonso J. Rodriguez-Morales.

**Data curation:** Darwin A. León-Figueroa, Joshuan J. Barboza, Ranjit Sah, Mario J. Valladares-Garrido, Alfonso J. Rodriguez-Morales.

**Formal analysis:** Darwin A. León-Figueroa, Abdelmonem Siddiq, Ranjit Sah, Mario J. Valladares-Garrido.

**Investigation:** Darwin A. León-Figueroa, Joshuan J. Barboza, Abdelmonem Siddiq, Ranjit Sah, Mario J. Valladares-Garrido, Alfonso J. Rodriguez-Morales.

**Methodology:** Darwin A. León-Figueroa, Joshuan J. Barboza, Abdelmonem Siddiq, Ranjit Sah, Mario J. Valladares-Garrido, Alfonso J. Rodriguez-Morales.

**Project administration:** Darwin A. León-Figueroa, Mario J. Valladares-Garrido, Alfonso J. Rodriguez-Morales.

**Resources:** Darwin A. León-Figueroa, Joshuan J. Barboza, Mario J. Valladares-Garrido, Alfonso J. Rodriguez-Morales.

**Software:** Darwin A. León-Figueroa, Mario J. Valladares-Garrido.

**Supervision:** Darwin A. León-Figueroa, Abdelmonem Siddiq, Ranjit Sah, Mario J. Valladares-Garrido, Alfonso J. Rodriguez-Morales.

**Validation:** Darwin A. León-Figueroa, Ranjit Sah, Mario J. Valladares-Garrido.

**Visualization:** Darwin A. León-Figueroa, Joshuan J. Barboza, Abdelmonem Siddiq, Mario J. Valladares-Garrido, Alfonso J. Rodriguez-Morales.

**Writing – original draft:** Darwin A. León-Figueroa, Joshuan J. Barboza, Abdelmonem Siddiq, Ranjit Sah, Mario J. Valladares-Garrido, Alfonso J. Rodriguez-Morales.

**Writing – review & editing:** Darwin A. León-Figueroa, Joshuan J. Barboza, Abdelmonem Siddiq, Ranjit Sah, Mario J. Valladares-Garrido, Alfonso J. Rodriguez-Morales.

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
