## [Decision Letter · Decision Letter 0]

14 Jun 2024

PONE-D-24-09033Knowledge and Attitude towards Monkeypox: Systematic review and meta-analysisPLOS ONE

Dear Dr. Valladares-Garrido,

Thank you for submitting your manuscript to PLOS ONE. After careful consideration, we feel that it has merit but does not fully meet PLOS ONE’s publication criteria as it currently stands. Therefore, we invite you to submit a revised version of the manuscript that addresses the points raised during the review process.

We look forward to receiving your revised manuscript.

Kind regards,

Mahmoud Kandeel

Academic Editor

PLOS ONE

Journal Requirements:

3. Please include your tables as part of your main manuscript and remove the individual files. Please note that supplementary tables (should remain/ be uploaded) as separate ""supporting information"" files"

Reviewers' comments:

Reviewer's Responses to Questions

**Comments to the Author**

1. Is the manuscript technically sound, and do the data support the conclusions?

Reviewer #1: Yes

Reviewer #2: No

Reviewer #3: Yes

Reviewer #4: Yes

2. Has the statistical analysis been performed appropriately and rigorously? 

Reviewer #1: Yes

Reviewer #2: No

Reviewer #3: Yes

Reviewer #4: Yes

3. Have the authors made all data underlying the findings in their manuscript fully available?

Reviewer #1: Yes

Reviewer #2: Yes

Reviewer #3: Yes

Reviewer #4: Yes

4. Is the manuscript presented in an intelligible fashion and written in standard English?

Reviewer #1: Yes

Reviewer #2: Yes

Reviewer #3: Yes

Reviewer #4: Yes

5. Review Comments to the Author

Reviewer #1: Summary

In this paper, the authors conducted a systematic review and meta-analysis to determine knowledge and attitudes towards Monkey pox and pose a good argument for justifying the study. The methods used in conducting the review are clearly described and are robust as they conform to the PRISMA checklist and the use of three independent reviewers. The authors report suboptimal prevalence of good knowledge and positive attitude toward Mpox and advocate for an increase in education and communication to improve them.

I recommend that this paper be accepted after the following revisions:

Major Issues

Discussion

1) The discussion section needs some revision. Most comparisons made with the pooled estimates of the review were from the individual studies that were included in the systematic review. I suggest that the authors focus on the overall implications of the results and discuss aspects of validity or quality of the evidence, biases and strengths and limitations of the paper.

2) Discuss the implications of the possible presence of publication bias as mentioned under section 3.3 Quality of included studies and publication bias.

Minor Issue

3) Exclusion criteria: “Lastly, attempts were made to establish contact with the corresponding author via e-mail, but unfortunately, it was impossible”. Please revise the statement as it is unclear. Is reference being made to articles lacking full text?

Reviewer #2: The meta-analysis presents significant heterogeneity (I² > 90%) between the included studies, persisting even after performing subgroup analyses. This high level of heterogeneity raises concerns about the reliability and validity of pooled prevalence estimates. The manuscript does not adequately address the underlying causes of this heterogeneity nor does it discuss its implications for the generalization of results. The discussion section inadequately addresses the issue of persistent heterogeneity, leading to a superficial interpretation of the results and an unclear understanding of how this variability affects the overall conclusions. An in-depth discussion of heterogeneity is essential, including possible reasons for it and how it affects the interpretation of results. The discussion should encompass considerations of methodological differences, cultural contexts, and the temporality of included studies to ensure better interpretation of results.

Reviewer #3: Dear authors: The manuscript is written well and the subject is very important in the present scenario. Increased ecological and temporal gap and decreased small pox vaccine protected population, the number of outbreak of monkey pox have increased. The specific knowledge of certain transmission factor are very important for prevention and control of the monkey pox and inclusion of specific knowledge parameter in the manuscript would increase the impact of the study.

1. In subheading of Knowledge about monkeypox and Attitude towards monkey pox which are main parameter of the study, more detail should be included as it is not clearly mentioned. The criteria on which you have determined the pooled prevalence of knowledge (knowledge about modes of transmission, clinical symptoms, treatment and prevention etc) and attitude should have been mentioned in the manuscript.

2. To perform this evaluation, a checklist composed of eight critical parameters was used to assess the responses as "yes," "no," "unclear," or "not applicable.": give the detail of these eight parameters and one parameter is “Dealing with confounding factors”, and in the selected studies there was no consideration of confounding variables so why it was included in scoring.

3. A forest plot format was used to present the pooled prevalence of monkeypox and attitudes towards it, including 95% confidence intervals: in this sentence are you saying the pooled prevalence of monkey pox or pooled prevalence of knowledge of monkey pox, please check it and correct it

4. Of the included studies, 27 addressed the prevalence of knowledge, six addressed the prevalence of attitude, and six managed both knowledge and attitude towards monkeypox: in this sentence

27 addressed the prevalence of knowledge

Six addressed the prevalence of attitude

Six managed both knowledge and attitude towards monkeypox

Total no of article taken is not clear ?

5. As you included Limitations of the study is the absence of a standardized measurement of "good knowledge" and "positive attitude" among different population groups in the published research, but some common parameter used to determine the knowledge and attitude in these studies can be discussed and /or could have been analyzed, so the impact of this research would be increased substantially.

a. Please explain what positive attitude is and how it may affect the control of monkey pox in discussion.

b. The inclusion of parameters used to assess the knowledge of monkey pox in the studied articles would be more helpful in the process of making strategies, as the knowledge and awareness certainly play major role in prevention and control of the diseases.

Reviewer #4: The authors' efforts on a meticulously executed systematic review are commendable. The comprehensive analysis and clear presentation greatly enhance understanding of the topic.

1. Short explaination about Inverse variance weighted random effects model and Egger's regression test's applications and interpretation can be added.

2. Couldnt find results from funnel plot in article.

3. In the supplementary material, subgroup analysis was provided according to study year, whereas in the article, it is mentioned as subgroup analysis based on the year of publication. Does this imply that the study year and publication year are the same for all the studies?

6. PLOS authors have the option to publish the peer review history of their article (what does this mean?). If published, this will include your full peer review and any attached files.

Reviewer #1: No

Reviewer #2: **Yes: **Andressa Souza Cardoso

Reviewer #3: **Yes: **Baleshwari Dixit

Reviewer #4: No

---

## [Author Response · Author response to Decision Letter 0]

4 Jul 2024

Dear Editor,

Thank you very much for reviewing our article, " Knowledge and Attitude towards Monkeypox: Systematic review and meta-analysis" Your suggestions and comments will be addressed below. Thank you for your valuable time and excellent review.

Editor's comments

Our response: 

Thank you for providing us with the reviewers' comments. We would like to inform you that each comment and suggestion has been addressed in detail.

The entire article has been reviewed.

The English grammar has been checked.

The format of the article has been adjusted according to the instructions for authors.

The data on monkeypox have been updated.

Each reviewer's comment has been responded to.

The quality of the images has been evaluated.

The discussion has been restructured according to the reviewers' comments.

Reviewer #1: 

1. Reviewer says: “In this paper, the authors conducted a systematic review and meta-analysis to determine knowledge and attitudes towards Monkey pox and pose a good argument for justifying the study. The methods used in conducting the review are clearly described and are robust as they conform to the PRISMA checklist and the use of three independent reviewers. The authors report suboptimal prevalence of good knowledge and positive attitude toward Mpox and advocate for an increase in education and communication to improve them. I recommend that this paper be accepted after the following revisions:”

Our response: “Thank you very much for your review. Recommendations and comments will be addressed below.”

2. Reviewer says: “1) The discussion section needs some revision. Most comparisons made with the pooled estimates of the review were from the individual studies that were included in the systematic review. I suggest that the authors focus on the overall implications of the results and discuss aspects of validity or quality of the evidence, biases and strengths and limitations of the paper”

Our response: “Thank you very much for your suggestion. We have eliminated the comparison of individual studies and focused on comparing them with systematic review studies and meta-analyses. In addition, we focused on explaining the variations in each result as well as improving the identification of limitations and strengths. ”

3. Reviewer says: “2) Discuss the implications of the possible presence of publication bias as mentioned under section 3.3 Quality of included studies and publication bias.”

Our response: “Implications for the possible presence of publication bias were discussed. In addition, these considerations were included in the limitations section, and a detailed explanation was provided for each variation observed.”

4. Reviewer says: “3) Exclusion criteria: “Lastly, attempts were made to establish contact with the corresponding author via e-mail, but unfortunately, it was impossible”. Please revise the statement as it is unclear. Is reference being made to articles lacking full text? ”

Our response: “Thank you very much for your comment. The paragraph has been deleted so as not to create confusion. ”

Reviewer #2: 

1. Reviewer says: “The meta-analysis presents significant heterogeneity (I² > 90%) between the included studies, persisting even after performing subgroup analyses.”

Our response: “We agree with you, doctor. However, we proceeded to explain the variations in our results compared to other studies. In addition, we indicated the limitations and strengths. ”

2. Reviewer says: “This high level of heterogeneity raises concerns about the reliability and validity of pooled prevalence estimates. ”

Our response: “Although the results are highly heterogeneous, the methodology followed in the study, the statistical tests, the process of preparing quality research, and, above all, the possible factors that may have influenced the variability of the results should be considered. ”

3. Reviewer says: “The manuscript does not adequately address the underlying causes of this heterogeneity nor does it discuss its implications for the generalization of results.”

Our response: “Their comments are gratefully acknowledged. The discussion has been modified to focus on the findings and factors that may have influenced the variability of the results. The comparison of individual studies has been eliminated, and we focus on the suggestions provided.”

4. Reviewer says: “The discussion section inadequately addresses the issue of persistent heterogeneity, leading to a superficial interpretation of the results and an unclear understanding of how this variability affects the overall conclusions. ”

Our response: “Their observation has been resolved, and the factors that would allow for variability in the results have been indicated. ”

5. Reviewer says: “An in-depth discussion of heterogeneity is essential, including possible reasons for it and how it affects the interpretation of results.”

Our response: “Your comment has been added. ”

6. Reviewer says: “The discussion should encompass considerations of methodological differences, cultural contexts, and the temporality of included studies to ensure better interpretation of results.”

Our response: “Your comment has been added. ”

Reviewer #3: 

1. Reviewer says: “Dear authors: The manuscript is written well and the subject is very important in the present scenario. Increased ecological and temporal gap and decreased small pox vaccine protected population, the number of outbreak of monkey pox have increased. The specific knowledge of certain transmission factor are very important for prevention and control of the monkey pox and inclusion of specific knowledge parameter in the manuscript would increase the impact of the study. ”

Our response: “Thank you very much for your comment and review; it has allowed us to improve the quality of the article. ”

2. Reviewer says: “1. In subheading of Knowledge about monkeypox and Attitude towards monkey pox which are main parameter of the study, more detail should be included as it is not clearly mentioned. The criteria on which you have determined the pooled prevalence of knowledge (knowledge about modes of transmission, clinical symptoms, treatment and prevention etc) and attitude should have been mentioned in the manuscript. ”

Our response: “Your suggestion was incorporated in the methodology section. Our study aims to comprehensively assess knowledge about monkeypox. The criteria used to determine the combined prevalence of knowledge covered modes of transmission, clinical symptoms, treatment, prevention, and the diagnosis of monkeypox.

The suggestion to extract data on each indicator of monkeypox knowledge departs from our research objective. ”

3. Reviewer says: “2. To perform this evaluation, a checklist composed of eight critical parameters was used to assess the responses as "yes," "no," "unclear," or "not applicable.": give the detail of these eight parameters and one parameter is “Dealing with confounding factors”, and in the selected studies there was no consideration of confounding variables so why it was included in scoring. ”

Our response: “Thank you very much for your comment. We have corrected that section, resulting in a moderate score for the quality of the studies. You can corroborate this information in the supplementary material.”

4. Reviewer says: “3. A forest plot format was used to present the pooled prevalence of monkeypox and attitudes towards it, including 95% confidence intervals: in this sentence are you saying the pooled prevalence of monkey pox or pooled prevalence of knowledge of monkey pox, please check it and correct it ”

Our response: “Corrections were made in accordance with your recommendation.”

5. Reviewer says: “4. Of the included studies, 27 addressed the prevalence of knowledge, six addressed the prevalence of attitude, and six managed both knowledge and attitude towards monkeypox: in this sentence

27 addressed the prevalence of knowledge

Six addressed the prevalence of attitude

Six managed both knowledge and attitude towards monkeypox

Total no of article taken is not clear ? ”

Our response: “It was decided to correct and delete this information because it was confusing.”

6. Reviewer says: “5. As you included Limitations of the study is the absence of a standardized measurement of "good knowledge" and "positive attitude" among different population groups in the published research, but some common parameter used to determine the knowledge and attitude in these studies can be discussed and /or could have been analyzed, so the impact of this research would be increased substantially.

Our response: Thank you very much. Your suggestion was incorporated into the methodology, and the limitations section of the study was improved.

a. Please explain what positive attitude is and how it may affect the control of monkey pox in discussion.

Our response: “The definition of a positive attitude towards monkeypox was added to the methodology section. Thank you very much for your suggestion.”

b. The inclusion of parameters used to assess the knowledge of monkey pox in the studied articles would be more helpful in the process of making strategies, as the knowledge and awareness certainly play major role in prevention and control of the diseases. ”

Our response: “Thank you very much. Your suggestion was incorporated into the methodology and discussed the variation of results. ”

Reviewer #4: 

1. Reviewer says: “The authors' efforts on a meticulously executed systematic review are commendable. The comprehensive analysis and clear presentation greatly enhance understanding of the topic.”

Our response: “Thank you very much for your recommendations and comments.”

2. Reviewer says: “1. Short explanation about Inverse variance weighted random effects model and Egger's regression test's applications and interpretation can be added. ”

Our response: “A brief explanation was added in the methodology section on the inverse variance-weighted random effects model and the applications and interpretation of Egger's regression test.”

3. Reviewer says: “2. Couldn’t find results from funnel plot in article. ”

Our response: “An explanation was added to the supplementary material funnel chart. Supported by Egger's test.”

4. Reviewer says: “3. In the supplementary material, subgroup analysis was provided according to study year, whereas in the article, it is mentioned as subgroup analysis based on the year of publication. Does this imply that the study year and publication year are the same for all the studies? ”

Our response: “The section of the meta-analysis of subgroups by year of publication was corrected and removed, as most of the studies were conducted and published in different years. Most of the studies belong to the 2022 run.”

Sincerely, 

Mario J. Valladares-Garrido 

Universidad Continental, Lima 15046, Peru; mvalladares@continental.edu.pe

---

## [Decision Letter · Decision Letter 1]

25 Jul 2024

Knowledge and Attitude towards Monkeypox: Systematic review and meta-analysis

PONE-D-24-09033R1

Dear Dr. Valladares-Garrido,

We’re pleased to inform you that your manuscript has been judged scientifically suitable for publication and will be formally accepted for publication once it meets all outstanding technical requirements.

Kind regards,

Sirwan Khalid Ahmed

Academic Editor

PLOS ONE

Additional Editor Comments (optional):

Reviewers' comments:

Reviewer's Responses to Questions

**Comments to the Author**

1. If the authors have adequately addressed your comments raised in a previous round of review and you feel that this manuscript is now acceptable for publication, you may indicate that here to bypass the “Comments to the Author” section, enter your conflict of interest statement in the “Confidential to Editor” section, and submit your "Accept" recommendation.

Reviewer #1: All comments have been addressed

Reviewer #3: All comments have been addressed

Reviewer #4: All comments have been addressed

2. Is the manuscript technically sound, and do the data support the conclusions?

Reviewer #1: Yes

Reviewer #3: Yes

Reviewer #4: Yes

3. Has the statistical analysis been performed appropriately and rigorously? 

Reviewer #1: Yes

Reviewer #3: Yes

Reviewer #4: Yes

4. Have the authors made all data underlying the findings in their manuscript fully available?

Reviewer #1: Yes

Reviewer #3: Yes

Reviewer #4: Yes

5. Is the manuscript presented in an intelligible fashion and written in standard English?

Reviewer #1: Yes

Reviewer #3: Yes

Reviewer #4: Yes

6. Review Comments to the Author

Reviewer #1: (No Response)

Reviewer #3: Dear Authors: Yours efforts in compilation and presentation of the study are great and all the suggested point have been addressed.

Reviewer #4: (No Response)

7. PLOS authors have the option to publish the peer review history of their article (what does this mean?). If published, this will include your full peer review and any attached files.

Reviewer #1: No

Reviewer #3: **Yes: **Baleshwari Dixit

Reviewer #4: No

---

## [Editor Report · Acceptance letter]

1 Aug 2024

PONE-D-24-09033R1 

PLOS ONE

Dear Dr. Valladares-Garrido, 

I'm pleased to inform you that your manuscript has been deemed suitable for publication in PLOS ONE. Congratulations! Your manuscript is now being handed over to our production team.

Kind regards, 

on behalf of

Dr. Sirwan Khalid Ahmed 

Academic Editor

PLOS ONE